# Continual Learning: Applications and the Road Forward

**Eli Verwimp**[*]                    *KU Leuven, Belgium*
**Rahaf Aljundi**                    *Toyota Motor Europe, Belgium*
**Shai Ben-David**    *University of Waterloo, and Vector Institute, Ontario, Canada*
**Matthias Bethge**                    *University of Tübingen, Germany*
**Andrea Cossu**                    *University of Pisa, Italy*
**Alexander Gepperth**    *University of Applied Sciences Fulda, Germany*
**Tyler L. Hayes**                    *NAVER LABS Europe, France*
**Eyke Hüllermeier**                    *University of Munich (LMU), Germany*
**Christopher Kanan**    *University of Rochester, Rochester, NY, USA*
**Dhireesha Kudithipudi**    *University of Texas at San Antonio, TX, USA*
**Christoph H. Lampert**    *Institute of Science and Technology Austria (ISTA)*
**Martin Mundt**                    *TU Darmstadt & hessian.AI, Germany*
**Razvan Pascanu**                    *Google DeepMind, UK*
**Adrian Popescu**                    *Université Paris-Saclay, CEA, LIST, France*
**Andreas S. Tolias**    *Baylor College of Medicine, Houston, TX, USA*
**Joost van de Weijer**    *Computer Vision Center, UAB, Barcelona, Spain*
**Bing Liu**                    *University of Illinois at Chicago, USA*
**Vincenzo Lomonaco**                    *University of Pisa, Italy*
**Tinne Tuytelaars**                    *KU Leuven, Belgium*
**Gido M. van de Ven**                    *KU Leuven, Belgium*

**Reviewed on OpenReview:** `https://openreview.net/forum?id=axBIMcGZn9`

## Abstract

Continual learning is a subfield of machine learning, which aims to allow machine learning models to continuously learn on new data, by accumulating knowledge without forgetting what was learned in the past. In this work, we take a step back, and ask: "*Why should one care about continual learning in the first place?*". We set the stage by examining recent continual learning papers published at four major machine learning conferences, and show that memory-constrained settings dominate the field. Then, we discuss five open problems in machine learning, and even though they might seem unrelated to continual learning at first sight, we show that continual learning will inevitably be part of their solution. These problems are model editing, personalization and specialization, on-device learning, faster (re-)training and reinforcement learning. Finally, by comparing the desiderata from these unsolved problems and the current assumptions in continual learning, we highlight and discuss four future directions for continual learning research. We hope that this work offers an interesting perspective on the future of continual learning, while displaying its potential value and the paths we have to pursue in order to make it successful. This work is the result of the many discussions the authors had at the Dagstuhl seminar on Deep Continual Learning, in March 2023.

---

[*]Corresponding author: `eli.verwimp@kuleuven.be`

## 1   Introduction

Continual learning, sometimes referred to as lifelong learning or incremental learning, is a subfield of machine learning that focuses on the challenging problem of incrementally training models on a stream of data with the aim of accumulating knowledge over time. This setting calls for algorithms that can learn new skills with minimal forgetting of what they had learned previously, transfer knowledge across tasks, and smoothly adapt to new circumstances when needed. This is in contrast with the traditional setting of machine learning, which typically builds on the premise that all data, both for training and testing, are sampled i.i.d. (independent and identically distributed) from a single, stationary data distribution.

Deep learning models in particular are in need of continual learning capabilities. A first reason for this is their strong dependence on data. When trained on a stream of data whose underlying distribution changes over time, deep learning models tend to adapt to the most recent data, thereby "catastrophically" forgetting the information that had been learned earlier (French, 1999). Secondly, continual learning capabilities could reduce the very long training times of deep learning models. When new data are available, current industry practice is to retrain a model fully from scratch on all, past and new, data (see Example 3.4). Such retraining is time inefficient, sub-optimal and unsustainable, with recent large models exceeding 10 000 GPU days of training (Radford et al., 2021). Simple solutions, like freezing feature extractor layers, are often not an option as the power of deep learning hinges on the representations learned by those layers (Bengio et al., 2013). To work well in challenging applications in e.g. computer vision and natural language processing, they often need to be changed.

The paragraph above describes two naive approaches to the continual learning problem. The first one, incrementally training – or finetuning – a model only on the new data, usually suffers from suboptimal performance when models adapt too strongly to the new data. The second approach, repeatedly retraining a model on all data used so far, is undesirable due to its high computational and memory costs. The goal of continual learning is to find approaches that have a better trade-off between performance and efficiency (e.g. compute and memory) than these two naive ones. In the contemporary continual learning literature, this trade-off typically manifests itself by limiting memory capacity and optimizing performance under this constraint. Computational costs are not often considered in the current continual learning literature, although this is challenged in some recent works, which we discuss in Sections 2 and 4.1.

In this article, we highlight several practical problems in which there is an inevitable continual learning component, often because there is some form of new data available for a model to train on. We discuss how these problems require continual learning, and how in these problems that what is constrained and that what is optimized differs. Constraints are hard limits set by the environment of the problem (e.g. small devices have limited memory), under which other aspects, such as computational cost and performance, need to be optimized. Progress in the problems we discuss goes hand in hand with progress in continual learning, and we hope that they serve as a motivation to continue working on continual learning, and offer an alternative way to look at it and its benefits. Similarly, they can offer an opportunity to align currently common assumptions that stem from the benchmarks we use, with those derived from the problems we aim to solve. Section 3 describes some of these problems and in Section 4 we discuss some exciting future research directions in continual learning, by comparing the desiderata of the discussed problems and contemporary continual learning methods. The aim of this paper is to offer a perspective on continual learning and its future. We do not aim to cover the latest technical progress in the development of continual learning algorithms, for this we refer to e.g. Masana et al. (2022); Zhou et al. (2023); Wang et al. (2023).

## 2   Current continual learning

Before exploring different problem settings in which we foresee continual learning as a useful tool, we first wish to understand the current landscape. Our aim is to paint a clear picture of how memory and computational cost are generally approached in current continual learning papers. To achieve this, we examined continual learning papers accepted at four top machine learning conferences (ECCV '22, NeurIPS '22, CVPR '23 and ICML '23) to have a representative sample of the current field. We considered all papers with either *'incremental', 'continual', 'forgetting', 'lifelong'* or *'catastrophic'* in their titles, disregarding false positives.

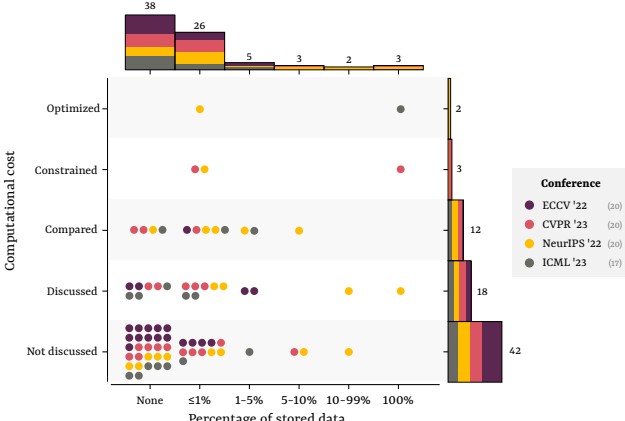

Figure 1: **Most papers strongly restrict data storage and do not discuss computational cost**. Each dot represents one paper, illustrating what percentage of data their methods store (horizontal axis) and how computational cost is handled (vertical axis). The majority of these papers are in the lower-left corner: those that strongly restrict memory use and do not quantitatively approach computational cost (i.e. it is at most discussed). For more details, see Appendix.

See Appendix for the methodology. For our final set of 77 papers, we investigated how they balance the memory and compute cost trade-offs. We discern five categories:

*Not discussed*: No clear mention of the impact of the proposed method/analysis on the cost.

*Discussed*: Cost is discussed in text, but not quantitatively compared between methods.

*Compared*: Cost is qualitatively compared to other methods.

*Constrained*: Methods are compared using the same limited cost.

*Optimized*: Cost is among the optimized metrics.

Many continual learning papers use memory in a variety of ways, most often in the form of storing samples, but regularly model copies (e.g. for distillation) or class means and their variances are stored as well. We focus on the amount of stored data, as this is the most common use of memory, but discuss other memory costs in the Appendix. Of the examined papers, all but three constrain the amount of stored samples. So rather than reporting the category, in Figure 1, we report how strongly data storage is constrained, using the percentage of all data that is stored. It is apparent that the majority of these papers do not store any (raw) samples and many store only a small fraction. Three notable exceptions that store all the raw data are a paper on continual reinforcement learning (RL) (Fu et al., 2022), something which is not uncommon in RL, see Section 3.5. The second one, by Prabhu et al. (2023a), studies common CL algorithms under a restricted computational cost. Finally, Kishore et al. (2023) study incremental document retrieval, a practical problem involving continual learning.

While memory costs (for raw samples) are almost always constrained, computational costs are much less so. Sometimes simply discussing that there is (almost) no additional computational cost can suffice, especially in settings where memory is the crucial limiting factor. Yet it is remarkable that in more than 50% of the papers there is no mention of the computational cost at all. When it is compared, it is often done in the appendix. There are a few notable exceptions among the analyzed papers that focus explicitly on the influence of the computational cost, either by constraining (Prabhu et al., 2023a; Kumari et al., 2022; Ghunaim et al., 2023) or optimizing it (Wang et al., 2022b; Kishore et al., 2023). For a more elaborate discussion of measuring the computational cost, see Section 4.1. Together, these results show that many continual learning methods are developed with a low memory constraint, and with limited attention to the computational cost. They are two among other relevant dimensions of continual learning in biological systems (Kudithipudi et al., 2022) and artificial variants (Mundt et al., 2022), yet with the naive solutions of the introduction in mind, they

are two crucial components of any continual learning algorithm. In the next section, we introduce some problems for which continual learning is inevitable. They illustrate that methods with a low computational cost are, just like methods that work in memory restricted settings, an important setting, yet they have not received the same level of attention.

## 3 Continual learning is not a choice

To solve the problems described in this section, continual learning is necessary and not just a tool that one could use. We argue that in all of them, the problem can, at least partly, be recast as a continual learning problem. This means that the need for continual learning algorithms arises from the nature of the problem itself, and not just from the choice of a specific way for solving it. We start these subsections by explaining what the problem is and why it fundamentally requires continual learning. Next, we briefly discuss current solutions and how they relate to established continual learning algorithms. We conclude each part by laying down what the constraints are and what metrics should be optimized.

### 3.1 Model editing

It is often necessary to correct wrongly learned predictions from past data. Real world practice shows us that models are often imperfect, e.g. models frequently learn various forms of decision shortcuts (Lapuschkin et al., 2019), or sometimes the original training data become outdated and are no longer aligned with current facts (e.g. a change in government leaders). Additionally, strictly accumulating knowledge may not always be compliant with present legal regulations and social desiderata. Overcoming existing biases, more accurately reflecting fairness criteria, or adhering to privacy protection regulations (e.g. the right to be forgotten of the GDPR in Europe (European Union, 2016)), represent a second facet of this editing problem.

When mistakes are exposed, it is desirable to selectively edit the model without forgetting other relevant knowledge and without re-training from scratch. Such edits should thus only change the output of a model to inputs in the neighborhood of the effected input-output mapping, while keeping all others constant. The model editing pipeline (Mitchell et al., 2022) first identifies corner cases and failures, then prompts data collection over those cases, and subsequently retrains/updates the model. Recently proposed methods are able to locally change models, yet this comes at a significant cost, or model draw-down, i.e. forgetting of knowledge that was correct (Santurkar et al., 2021). Often the goal of model editing is to change the output associated with a specific input from A to B, yet changing the output to something generic or undefined is an equally interesting case. Such changes can be important in privacy-sensitive applications, to e.g. forget learned faces or other personal attributes.

Naively, one could retrain a model from scratch with an updated dataset, that no longer contains outdated facts and references to privacy-sensitive subjects, or includes more data on previously out-of-distribution data. To fully retrain on the new dataset, significant computational power and access to all previous training data is necessary. Instead, with effective continual learning, this naive approach can be improved by only changing what should be changed. An ideal solution would be able to continually fix mistakes, at a much lower computational cost than retraining from scratch, without forgetting previously learned and unaffected information. Such a solution would minimize computational cost, while maximizing performance. There is no inherent limitation on memory in this problem, although it can be limited if not all training data are freely accessible.

### 3.2 Personalization and specialization

Some of the most powerful machine learning models are trained on very large datasets, usually scraped from the Internet. The result is a model that is able to extract useful and diverse features from high-dimensional data. However, the vastness of the data they are trained on also has a downside. Internet data is generated by many different people, who all have their own preferences and interests. One model cannot fit these conflicting preferences, and the best fit is close to the average internet user (Hu et al., 2022b). However, machine learning models are often used by individuals or small groups, or for highly specific applications. This contradiction makes any possessive references such as 'my car' or 'my favorite band' by construction

---

**Example: 3.1**

Lazaridou et al. (2021) used the customnews benchmark to evaluate how well a language model trained on news data from $1969 - 2017$ performs on data from 2018 and 2019. They find that models perform worse on the newest data, mostly on proper nouns (e.g. "Ardern" or "Khashoggi"), as well as words introduced because of societal changes such as "Covid-19" and "MeToo". They identify a set of 287 new words that were not used in any document prior to 2018. Such new words are inevitable in future texts too. To teach a model these changes they perform updates on the newly arriving data, which gradually improves the performance on the years 2018 and 2019 (a 10% decrease in perplexity), yet at the cost of performance on earlier years (a 5% increase on *all* previous years). When weighing all years equally, the final model thus got worse than before updating. 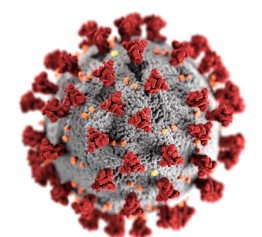

---

ambiguous and impossible for the system to understand. Further, Internet scraped data do not always contain (enough) information to reach the best performance in specialized application domains like science and user sentiment analysis (Beltagy et al., 2019). In contrast to Section 3.1, here there are many new data that have a strong relation to each other (e.g. scientific words), whereas model editing typically concerns fewer and less related, individual changes.

Domain adaptation and personalization are thus often necessary. The topic has been investigated in the natural language processing (NLP) community for many different applications. Initially, fine-tuning on a supervised domain-specific dataset was the method of choice, but recently, with the success of very large language models (LLM), the focus has shifted towards changing only a small subset of parameters with adapters (Houlsby et al., 2019), low-rank updates (Hu et al., 2022a) or prompting (Jung et al., 2023). However, these methods do not explicitly identify and preserve important knowledge in the original language model. This hampers the integration of general and domain-specific knowledge and produces weaker results (Ke et al., 2022). To identify the parameters that are important for the general knowledge in the LLM in order to protect them is a challenging problem. Recent works (Ke et al., 2021) made some progress in balancing the trade-off between performance on in-domain and older data. In the computer vision field, similar work has also been done by adapting CLIP to different domains (Wortsman et al., 2022) and to include personal text and image pairs (Cohen et al., 2022).

No matter how large or sophisticated the pre-trained models become, there will always be data that they are not, or cannot be, trained on (e.g. tomorrow's data). Specialized and personalized data can be collected afterwards, either by collecting new data or by extracting the relevant bits from the larger, original dataset. The addition of such a specialized dataset is a distribution shift, and thus continual learning algorithms are required. The final goal is to train a specialized or personalized model, more compute-efficient than when trained from scratch, without losing relevant performance on the pre-trained domain. This makes this problem different from transfer learning, where it is not a requirement (Zhuang et al., 2020). When training is done on the original server, past data are usually available and can be used to prevent forgetting. Yet sometimes this is not the case, because training happens on a more restricted (e.g. personal) device, then memory does become a constraint, which we elaborate on in the next subsection.

### 3.3 On-device learning

To offer an experience aligned with a user's preferences, or adjusted to a new personal environment, many deep learning applications require updates on the deployed device. Cloud computing is often not available because of communication issues (e.g. in remote locations with restricted internet access, or when dealing with very large quantities of data), or to preserve the privacy of the user (e.g. for domestic robots, monitoring cameras). On such small devices, both memory and computational resources are typically constrained, and the primary goal is to maximize model efficacy under these constraints. These tight constraints often make storing all user data and retraining from scratch infeasible, necessitating continual learning whenever the pre-trained capabilities should not be lost during continued on-device training (see also Example 3.3).

---

**Example: 3.2**

Dingliwal et al. (2023) personalize end-to-end speech recognition models with words that are personal to the user (e.g. family member names) or words that are very rare except in specialized environments (e.g. "ecchymoses" in medical settings). With an extra attention module and a precomputed set of representations of the specialized vocabulary, they 'bias' the original model towards using the new rare and unknown words. The performance on the specialized words is remarkably improved, yet with a decrease in performance on non-specialist word recognition. In their experiments specialized tokens are less than 1% off all tokens, so even a relatively small decrease in performance on other tokens is non-negligible.

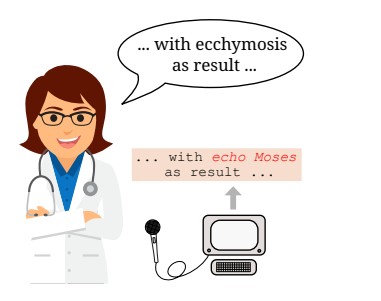

---

These constraints, as well as increasingly complex computations for energy-accuracy trade-offs in real-time (Kudithipudi et al., 2023), limit the direct application of optimization typically used in cloud deployments. For example, existing methods only update the final classification layer of a pre-trained feature extractor (Hayes & Kanan, 2022). Yet this relatively lightweight process becomes challenging when there is a large domain gap between the initial training set and the on-device data. The latter is often hard to collect, since labeling large amounts of data by the user is impractical, requiring few-shot solutions. When devices shrink even further, the communication costs become significant, and reading and writing to memory can be up to ∼99% of the total energy budget (Dally, 2022). In addition to algorithmic optimizations for continual learning, architectural optimizations offer interesting possibilities. These enhancements may include energy-efficient memory hierarchies, adaptable dataflow distribution, domain-specific compute optimizations like quantization and pruning, and hardware-software co-design techniques (Kudithipudi et al., 2022).

On-device learning from data that is collected locally almost certainly involves a distribution shift from the original (pre-)training data. This means the sampling process is no longer i.i.d., thus requiring continual learning to maintain good performance on the initial training set. If these devices operate on longer time scales, the data they sample themselves will not be i.i.d. either. To leverage the originally learned information as well as adapt to local distribution changes, such devices require continual learning to operate effectively. Importantly, they should be able to learn using only a limited amount of labeled information, while operating under the memory and compute constraints of the device.

---

**Example: 3.3**

In a 2022 article by MIT Review (Guo, 2022), it was revealed how a robot vacuum cleaner had sent images, in some cases sensitive ones, back to the company, to be labeled and used in further training on central servers. In response, an R&D director of the company stated: *"Road systems are quite standard, so for makers of self-driving cars, you'll know how the lane looks […], but each home interior is vastly different"*, acknowledging the need to adjust the robots to the environment they are working in. Our homes are highly diverse, but also one of the most intimate and private places that exist. Images can reveal every detail about them and should thus remain private. Adapting to individual homes is necessary, but should not come at the cost of initial smart abilities such as object recognition, collision prevention and planning, which are unlikely to be learned using only locally gathered data.

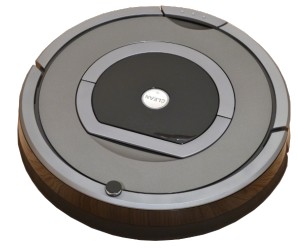

---

### 3.4 Faster retraining with warm starting

In many industrial settings, deep neural networks are periodically re-trained from scratch when new data are available, or when a distribution shift is detected. The newly gathered data is typically a lot smaller than the original dataset is, which makes starting from scratch a wasteful endeavor. As more and more data is collected, the computational requirements for retraining continue to grow over time. Instead, continual learning can start from the initial model and only update what is necessary to improve performance on the

new dataset. Most continual learning methods are not designed for computational efficiency (Harun et al., 2023a), yet Harun et al. (2023b) show that reductions in training time by an order of magnitude are possible, while reaching similar performance. Successful continual learning would offer a way to drastically reduce the expenses and extraordinary carbon footprint associated with retraining from scratch (Amodei & Hernandez, 2018), without sacrificing accuracy.

The challenge is to achieve performance equal to or better than a solution that is trained from scratch, but with fewer additional resources. One could say that it is the performance that is constrained, and computational cost that must be optimized. Simple approaches, like warm-starting, i.e. from a previously trained network, can yield poorer generalization than models trained from scratch on small datasets (Ash & Adams, 2020), yet it is unclear whether this translates to larger datasets, and remains a debated question. Similar results were found in (Berariu et al., 2021; Dohare et al., 2023), which report a loss of plasticity, i.e. the ability to learn new knowledge after an initial training phase. In curriculum learning (Bengio et al., 2009), recent works have tried to make learning more efficient by cleverly selecting which samples to train on when (Hacohen et al., 2020). Similarly, active learning (Settles, 2009) studies which unlabeled samples could best be labeled (given a restricted budget) to most effectively learn. Today those fields have to balance learning new information with preventing forgetting, yet with successful continual learning they could focus more on learning new information as well and as quickly as possible.

Minimizing computational cost could also be rephrased as maximizing learning efficiency. Not having to re-learn from scratch whenever new data is available, figuring out the best order to use data for learning, or the best samples to label can all contribute to this goal. Crucially, maximizing knowledge accumulation from the available data is part of this challenge. Previous work (Hadsell et al., 2020; Hacohen et al., 2020; Pliushch et al., 2022) suggested that even when all data is used together, features are learned in sequential order. Exploiting this order to make learning efficient requires continual learning.

---

**Example: 3.4**

*Continuous* training is one of the six important building blocks in MLOps (Machine Learning Operations, similar to DevOps), according to a Google white paper on the subject (Salama et al., 2021). This step is considered necessary, in response to performance decays when incoming data characteristics change. They describe in great detail how to optimize this pipeline, from various ways to trigger retraining to automated approaches to deploy retrained models. However, retraining is implicitly considered to be from scratch, which makes most pipelines inherently inefficient. Similarly, other resources stating the importance of retraining ML models and efficient MLOps, at most very briefly consider other options than retraining from scratch (Kreuzberger et al., 2023; Komolafe, 2023; Alla et al., 2021). The efficiency that can be gained here represents an enormous opportunity for the continual learning field, which is clearly illustrated by Huyen (2022) from an industry perspective.

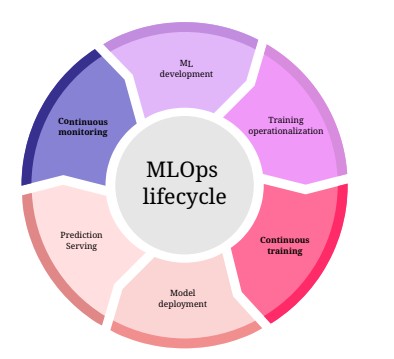

---

### 3.5 Reinforcement learning

In reinforcement learning (RL), agents learn by interacting with an environment. This creates a loop, where the agent takes an action within the environment, and receives from the environment an observation and a reward. The goal of the learning process is to learn a policy, i.e. a strategy to choose the next action based on the observations and rewards seen so far, which maximizes the rewards (Sutton & Barto, 2018). Given that observations and rewards are conditioned on the policy, this leads to a natural non-stationarity, where each improvement step done on the policy can lead the agent to explore new parts of the environment. The implicit non-stationarity of RL can be relaxed to a piece-wise stationary setting in off policy RL settings (Sutton & Barto, 2018), however this still implies a continual learning problem. Offline RL (Levine et al., 2020) (e.g. imitation learning) completely decouples the policy used to collect data from the learning policy, leading to a static data distribution, though is not always applicable and can lead to suboptimal solutions

due to the inability of the agent to explore. Lastly, for real-world problems, the environment itself may be non-stationary, either intrinsically so, or through the actions of the agent.

The presence of non-stationarities in reinforcement learning makes efficient learning difficult. To accelerate learning, experience replay has been an essential part of reinforcement learning (Lin, 1992; Mnih et al., 2015). While engaging in new observations, previously encountered states and action pairs are replayed to make training more i.i.d. In contrast to replay in supervised learning, in RL there is less focus on restricting the amount of stored examples, as the cost of obtaining them is considered very high. Instead the focus is how to select samples for replay (e.g. Schaul et al., 2016) and how to create new experiences from stored ones (Lin et al., 2021). Additionally, loss of plasticity (e.g. Dohare et al., 2023; Lyle et al., 2022) — inability of learning efficiently new tasks — and formalizing the concept of continual learning (e.g. Kumar et al., 2023; Abel et al., 2023) also take a much more central role in the RL community.

Finally, besides the non-stationarities encountered while learning a single task, agents are often required to learn multiple tasks. This setting is an active area of research (Wołczyk et al., 2021; Kirkpatrick et al., 2017; Rolnick et al., 2019), particularly since the external imposed non-stationarity allows the experimenter to control it and probe different aspects of the learning process. RL has its own specific problems with continual learning, e.g. trivially applying rehearsal methods fails in the multi-task setting, and not all parts of the network should be regularized equally (Wolczyk et al., 2022). Issues considering the inability to learn continually versus the inability to explore an environment efficiently, as well as dealing with concepts like episodic and non-episodic RL, makes the study of continual learning in RL more challenging. Further research promises agents that train faster, learn multiple different tasks sequentially and effectively re-use knowledge from previous tasks to work faster and towards more complex goals.

---

**Example: 3.5**

Typical RL methods store millions or more transitions in a replay memory. Schaul et al. (2016) showed that theoretically exponential training speed-ups are possible when cleverly selecting the transitions to replay. By approximating 'how much the model can learn' from a transition, they prioritize some samples over others and practically show a linear speed-up compared to uniform selection, the default at that point. Current state-of-the-art in the Atari-57 benchmark, MuZero (Schrittwieser et al., 2020), relies on this prioritized selection and confirms its importance, yet from the initial theoretical results, it is clear that improved continual learning could further improve convergence speeds and results (e.g. Pritzel et al., 2017).

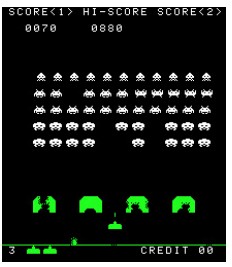

---

## 4 Future directions for continual learning

In this section we discuss interesting future directions for continual learning research, informed by what was discussed in the previous sections. We start by addressing the motivation for these research directions, followed by a brief overview of existing work, and finally justifying the importance of each concept.

### 4.1 Rethinking memory and compute assumptions

In all of the problems described in the previous section, optimizing or restricting compute complexity plays an important role, often a more central one than memory capacity does. This is in stark contrast to the results in Section 2. The vast majority of papers does not qualitatively approach compute complexity, while not storing, or only very few, samples. Two popular reasons for arguing a low storage solution are the cost of memory and privacy concerns, but these arguments are often not relevant in practice. Prabhu et al. (2023b) calculate that the price to store ImageNet1K for one month is just 66¢, while training a model on it requires 500$. This means storing the entire dataset for 63 years is as expensive as training ImageNet once. Further, privacy and copyright concerns are not solved by simply deleting data from the training set, as data can be recovered from derivative models (Haim et al., 2022), and rulings to remove data might only be viable by re-training from scratch (Zhao, 2022) (hence making continual learning superfluous), at least until reliable

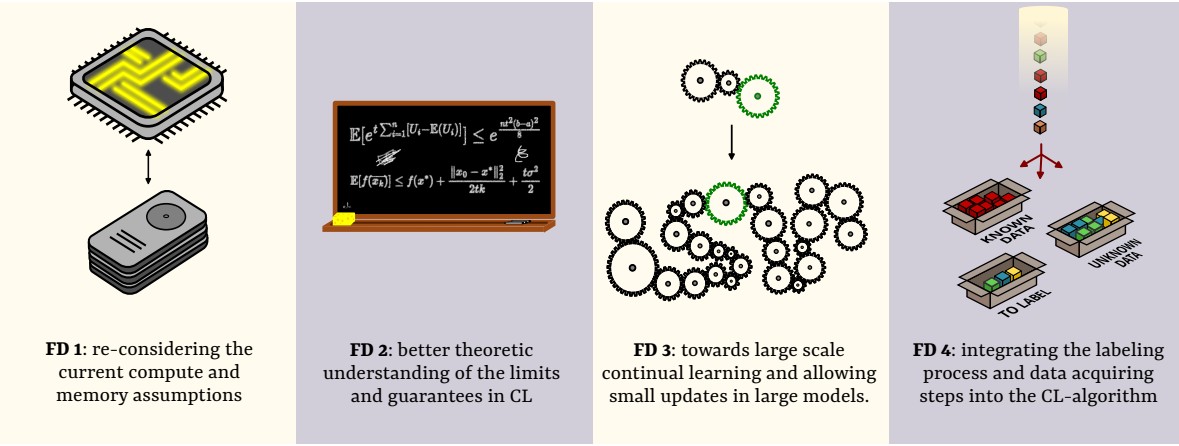

Figure 2: An overview of the future directions (FD) discussed in Section 4

model editing exists (see Section 3.1). Section 3.3 showed that use cases in low memory settings exist, but, as four of the five problems show, there are many reasons to study algorithms that restrict computational cost just like restricted memory settings are studied today. We believe it is important to reconsider these common assumptions on memory and computational cost, and instead derive them from the real-world problems that continual algorithms aim to solve.

To achieve this goal, we should agree on how to measure computational cost, which is necessary to restrict it. Yet it is not straightforward to do so. Recent approaches use the number of iterations (Prabhu et al., 2023a) and forward/backward passes (Kumari et al., 2022; Harun et al., 2023c), which works well if the used model is exactly the same, but cannot capture architectural differences that influence computational cost. Similarly, when the number of parameters is used (Wang et al., 2022a), more iterations or forward passes do not change the perceived cost. The number of floating point operations (FLOPs) is often used to measure computational cost in computer science, and is a promising candidate, yet is sometimes hard to measure accurately (Wang et al., 2022b). In practice, limiting the size of the memory buffer can reduce the computational cost; it can for example reduce the convergence time, but it does not offer guarantees. Directly controlling the computational cost, rather than by proxy, can alter which algorithms perform best, e.g. in RL this is shown in Javed et al. (2023). Additionally, time to convergence should also be considered, as faster convergence would also lower compute time Schwartz et al. (2020). To properly benchmark compute time and memory use in continual learning algorithms, we should build on this existing literature to attain strong standards for measuring both compute and memory cost and the improvements thereof.

As illustrated in Section 2, there are works that have started to question our common assumptions in continual learning (Harun et al., 2023b). SparCL optimizes compute time explicitly (Wang et al., 2022b), while (Prabhu et al., 2023b;a) compare methods while constraining computational cost. Chavan et al. (2023) establish DER (Yan et al., 2021) as a promising method when compute complexity is constrained, while other works suggest that experience replay is likely most efficient (Harun et al., 2023c; Prabhu et al., 2023a). These early works have laid the groundwork, and we believe that it is in continual learning's best interest to push further in this direction, and develop strategies for learning under a tight compute budget, with and especially *without* memory constraints.

## 4.2   Theory

In the past, continual learning research has achieved interesting empirical results. In contrast to classic machine learning, not much is known about whether and under which conditions, we can expect results. Many theoretical results rely on the i.i.d assumption, among which the convergence of stochastic gradient descent and the difference between expected and empirical risk in many PAC-bound analyses (although there are some exceptions, e.g. Pentina & Lampert 2014). Crucially, the i.i.d. assumption is almost always broken in continual learning, as illustrated by the problems in Section 3. To a certain extent this also happens

in training with very large datasets, due to the computational cost of sampling data batches in an i.i.d. fashion compared to ingesting them in fixed but random order. Not having theoretical guarantees means that continual learning research is often shooting in the dark, hoping to solve a problem that we do not know is solvable in the first place.

To understand when and under which assumptions we can find solutions, new concepts in a number of directions need to be developed in order to theoretically grasp continual learning in its full breadth. A key aspect is optimization. In which sense and under which conditions do continual learning algorithms converge to stable solutions? And what kind of generalization can we expect? We want to emphasize that we should not be misguided by classical notions of those concepts. It might be, for instance, more insightful to think of continual learning as tracking a time-varying target when reasoning about convergence (e.g. Abel et al., 2023). Algorithms designed from this perspective can outperform solutions that are converged in the classical sense, a result of acknowledging the temporal coherence of our world (Sutton et al., 2007; Silver et al., 2008). They show the potential for continual learning algorithms to improve over static systems, although that does not mean more classical, static notions of convergence are not useful, as illustrated by Zimin & Lampert (2019). Even if it is possible to find a good solution, it is unclear whether this is achievable in reasonable time, and crucially, whether it can be more efficient than re-training from scratch. Knoblauch et al. (2020) show that even in ideal settings continual learning is NP-complete, yet Mirzadeh et al. (2021) empirically illustrate that often there are linear low-loss paths to the solution, reassuring that solutions that are easy to find are not unlikely to exist.

Not all continual learning is equally difficult. An important factor is the relatedness of old and new data. In domain adaptation, David et al. (2010) have shown that without assumptions on the data, some adaptation tasks are simply impossible. Empirically (Zamir et al., 2018) and to some extent theoretically (Prado & Riddle, 2022), we know that in many cases transfer is successful because most tasks are related. Similar results for continual learning are scarce. Besides data similarity, the problem setting is an important second facet. For instance, class incremental learning is much harder than its task-incremental counterpart, as it additionally requires the predictions of task-identities (van de Ven et al., 2022; Kim et al., 2022). We believe that understanding the *difficulty of a problem* and having *formal tools expressive enough to describe or understand relatedness between natural data* will allow a more principled approach, and better guarantees on possible results.

Finally, theory in continual learning might simply be necessary to deploy continual learning models in a trustworthy manner. It requires models to be certified (Huang et al., 2020), i.e. they need to be thoroughly tested before deployment to work as intended. It is however unclear how this would fare in a continual learning setting, as by design, such models will be updated after deployment.

### 4.3 Large-scale continual learning

Most of the problems in Section 3 start when there is a change in the environment of the model and it needs to be updated. These changes are often are small compared to the preceding training. The initial models, often referred to as foundation models, are typically powerful generalist models that can perform well on various downstream tasks, e.g. Oquab et al. (2023); Radford et al. (2021). However, performance gains are generally seen when adapting these models to specific tasks or environments, which compromises the initial knowledge in the pretrained model. In a continuously evolving world, one would expect that this knowledge is subject to be continuous editing, updating, and expansion, without losses in performance. When investigating continual learning that starts with large-scale pretrained models, the challenges might differ from those encountered in continual learning from random initializations and smaller models.

In contrast to smaller models, the required adjustments to accommodate new tasks are usually limited compared to the initial training phase, which may result in forgetting being less pronounced than previously anticipated. It is an open questions which continual learning techniques are more effective in such a case. For example, Xiang et al. (2023) suggest that parameter regularization mechanisms (Kirkpatrick et al., 2017) are more effective than functional regularization (e.g. distillation approaches (Li & Hoiem, 2017)) in reducing forgetting in a large language model. Additionally, it might not be necessary to update all parameters, which in itself is computationally expensive for large models. Approaches considering adapters

(Houlsby et al., 2019; Jia et al., 2022; Li & Liang, 2021), low rank updates (Hu et al., 2022a) or prompting (Jung et al., 2023), are argued to be more feasible in this setting. Freezing, or using non-uniform learning rates, might be necessary when the amount of data is limited to prevent optimization and overfitting issues, or to improve the implicit bias of a model when faced with a new data (Sutton, 1992) How to adapt models if the required changes are comparatively small to the original training remains an interesting research direction, with promising initial results (Wang et al., 2022c; Li et al., 2023; Panos et al., 2023).

Lastly, in the large-scale learning setting there is a paradigm shift from end-to-end learning towards more modular approaches, where different components are first trained and then stitched together. It is somewhat of an open question of what implication this has for continual learning (Ostapenko et al., 2022; Cossu et al., 2022). In the simplest scenario, one could decouple the learning of a representation, done with e.g. contrastive unsupervised learning, versus that of classifier with supervision (e.g. Alayrac et al., 2022). Yet this idea can be extended towards using multiple (e.g. domain specific) experts (Ramesh & Chaudhari, 2021) and using more than one modality (e.g. vision and speech) (Radford et al., 2021). A better understanding of how continual learning algorithms can exploit these setting is required to expand beyond the end-to-end paradigms currently used.

While there has been promising research in these directions, we believe that considerably more is needed. So far, we do not have a strong understanding of the possibilities and limits of small updates on large pre-trained models, and how the training dynamics are different than the smaller-scale models typically used in continual learning. Further research in the relation between new data and pre-training data might open up new opportunities to more effectively apply these smaller updates, and will ultimately make continual learning more effective in handling all sorts of changes in data distributions. Understanding the interplay between memory and learning, and how to exploit the modular structure of this large model could enable specific ways to address the continual learning problem.

### 4.4 Continual learning in a real-world environment

Continual learning, in its predominant form, is centered around effective and resource-efficient accumulation of knowledge. The problem description typically starts whenever there is some form of new data available, see Section 3. How the data is produced is a question that is much less considered in continual learning. We want to emphasize that there is a considerable overlap between machine learning subfields (Mundt et al., 2022), in particular in those that are concerned with both detecting change in data distributions and techniques that reduce the required effort in labeling data. It will be important to develop continual learning algorithms with these in mind. These fields depend on and need each other to solve real-world problems, making it crucial that their desiderata align.

Open world learning (Bendale & Boult, 2015) is such a closely related field. Early work on open-world learning focused on detecting novel classes of objects that were not seen during training, relaxing the typical closed-world assumption in machine learning. A first step to realize open-world learning is detecting a change in incoming data, more formally known as out-of-distribution (OOD) or novelty detection. Detecting such changes requires a certain level of uncertainty-awareness of a model, i.e. it should quantify what it does and does not know. This uncertainty can be split into aleatoric uncertainty, which is an irreducible property of the data itself, and epistemic uncertainty, a result of the model not having learned enough (Hüllermeier & Waegeman, 2021). When modeled right, the latter can provide a valuable signal to identify what should be changed in continually trained models (Ebrahimi et al., 2020). Alternatively, it provides a theoretically grounded way for active learning, which studies how to select the most efficient unlabeled data points for labeling (Settles, 2009; Nguyen et al., 2022).

Even when OOD data is properly detected, it might not be directly usable. It can be unlabeled, without sufficient meta-data, or in the worst case corrupted. Many CL algorithms require the new data to be labeled before training, which is always costly and often difficult in e.g. on-device applications. This process makes it likely that when solving problems as described in Section 3, a model has access to a set of unlabeled data, possibly extended by some labeled samples that are obtained using active learning techniques. To successfully work in such an environment, a model should be able to update itself in a self- or semi-supervised way, an idea recently explored in Fini et al. (2022).

Continual learning depends on the data available to update a model. It is thus important to develop CL algorithms that are well calibrated, capable of OOD detection and learning in an open world. Further, in many settings (see Section 3.3), new data will not, or only partly, be labeled, which requires semi- or self-supervised continual learning (Mundt et al., 2023). We recommend working towards future continual learning algorithms with these considerations in mind, as methods that rely less on the fully labeled, closed-world assumption will likely be more practically usable in the future.

## 5 Conclusion

In this work, we first examined the current continual learning field, and showed that many papers study the memory-restricted setting with little or no concern for the computational cost. The problems we introduced all require some form of continual learning, not because it is a nice-to-have, but because the solution inherently depends on continual learning. Finally, we established four research directions in continual learning that we find promising, in the light of the scenarios we described. In summary, many of these applications are more compute-restricted than memory-restricted, so we vouch for exploring this setting more. Further, we believe a better theoretical understanding, a larger focus on pre-training and comparatively small future updates, and greater attention to how data is attained, will help us solving these problems, and make continual learning a practically useful tool to solve the described and other machine learning problems. A summary of the talks and discussions at the Deep Continual Learning seminar in Dagstuhl that inspired this paper can be found in Tuytelaars et al. (2023).

## Broader impact

This paper does not present any new algorithm or dataset, hence the potential *direct* societal and ethical implications are rather limited. However, continual learning and applications thereof, as we have examined, may have a long-term impact. Reducing computational cost can positively affect the environmental impact machine learning has. Easily editable networks, or ways to quickly update parts of networks as discussed in Section 3.1 and 3.4, may further democratize the training of machine learning model. Yet this also means that it can be exploited by malicious actors to purposely inject false information in a network. Predictions made by those networks could misinform people or lead to harmful decisions. Excessive personalization as described in Section 3.2 may negatively impact community solidarity, yet benefit the individual.

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

## A    Details of the analysis in Section 2

To verify the keywords *'incremental'*, *'continual'*, *'forgetting'*, *'lifelong'* and *'catastrophic'*, used to filter the papers based on their titles, we tested them using a manually collected validation set of which we are certain that they are continual learning related. This set was manually collected while doing research on continual learning over the past few years. The keywords were present in 96% of the paper titles. From each conference, we randomly picked up to 20 out of all matched papers, disregarding false positives.

It is common for to evaluate new methods and analyses on more than one benchmark. Often this means that the percentage of stored samples is not uniform across the experiments in a paper. In Figure 1, we showed the minimum percentage used, in Figure 3 we show the maximum. The conclusion remains the same, and the amount of stored samples is constrained in all but two benchmarks.

In Table 1 we provide a table of all the papers we used in the analysis of Section 2, showing their minimal and maximal sample store ratio (SSR) i.e. the percentage of samples stored, as well as possibly other memory consumption. The last column mentions how they approached the computational cost.

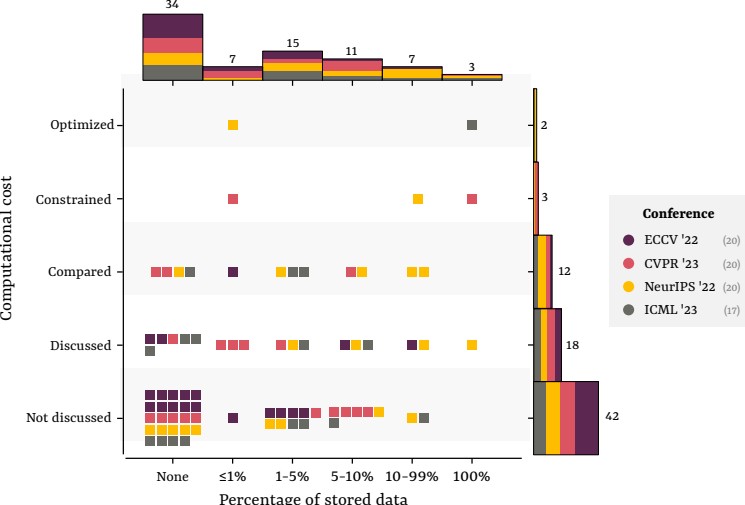

Figure 3: **Most papers strongly restrict memory use and do not discuss computational cost**. This figure is an alternate version of Figure 1, with the maximum percentage of stored samples rather than the minimum. Each dot represents one paper, illustrating what percentage of data their methods store (horizontal axis) and how computational complexity is handled (vertical axis). The majority of examined papers are in the lower-left corner: those that strongly restrict memory use and do not quantitatively approach computational cost

Table 1: All papers used in the examination of Section 2. SSR refers to the sample store ratio, i.e. how much samples are stored in relation to the entire dataset.

| # | Conference | Title | SSR (min) | SSR (max) | Memory (other) | Computational Cost |
|---|---|---|---|---|---|---|
| 1 | ECCV | Balancing Stability And Plasticity Through Advanced Null Space In Continual Learning | 0 | 0 | Null spaces | discussed |
| 2 | ECCV | Class-Incremental Novel Class Discovery | 0 | 0 | prototypes | not discussed |
| 3 | ECCV | Prototype-Guided Continual Adaptation For Class-Incremental Unsupervised Domain Adaptation | 0 | 0 | prototypes | not discussed |
| 4 | ECCV | Few-Shot Class-Incremental Learning Via Entropy-Regularized Data-Free Replay | 0 | 0 | generators | not discussed |
| 5 | ECCV | Anti-Retroactive Interference For Lifelong Learning | 0.02 | 0.2 | / | discussed |
| 6 | ECCV | Long-Tailed Class Incremental Learning | 0.01 | 0.04 | / | not discussed |
| 7 | ECCV | Dlcft: Deep Linear Continual Fine-Tuning For General Incremental Learning | 0.001 | 0.04 | / | not discussed |
| 8 | ECCV | Generative Negative Text Replay For Continual Vision-Language Pretraining | 0 | 0 | / | not discussed |
| 9 | ECCV | Online Continual Learning With Contrastive Vision Transformer | 0.001 | 0.02 | / | not discussed |
| 10 | ECCV | Coscl: Cooperation Of Small Continual Learners Is Stronger Than A Big One | 0 | 0.04 | generators | not discussed |
| 11 | ECCV | R-Dfcil: Relation-Guided Representation Learning For Data-Free Class Incremental Learning | 0 | 0 | generators | not discussed |
| 12 | ECCV | Continual Semantic Segmentation Via Structure Preserving And Projected Feature Alignment | 0 | 0 | / | discussed |
| 13 | ECCV | Balancing Between Forgetting And Acquisition In Incremental Subpopulation Learning | 0 | 0 | / | not discussed |
| 14 | ECCV | Few-Shot Class-Incremental Learning For 3d Point Cloud Objects | 0.001 | 0.001 | prototypes | not discussed |
| 15 | ECCV | Meta-Learning With Less Forgetting On Large-Scale Non-Stationary Task Distributions | 0.002 | 0.002 | model copies | compared |
| 16 | ECCV | Novel Class Discovery Without Forgetting | 0 | 0 | prototypes | not discussed |
| 17 | ECCV | Rbc: Rectifying The Biased Context In Continual Semantic Segmentation | 0 | 0 | model copies | not discussed |
| 18 | ECCV | Coarse-To-Fine Incremental Few-Shot Learning | 0 | 0 | / | not discussed |
| 19 | ECCV | Continual Variational Autoencoder Learning Via Online Cooperative Memorization | 0.02 | 0.1 | / | discussed |
| 20 | ECCV | Dualprompt: Complementary Prompting For Rehearsal-Free Continual Learning | 0 | 0 | prompts | not discussed |
| 21 | CVPR | Incrementer: Transformer for Class-Incremental Semantic Segmentation with Knowledge Distillation Focusing on Old Class | 0 | 0 | model copies | not discussed |
| 22 | CVPR | Real-Time Evaluation in Online Continual Learning: A New Hope | 0.001 | 0.001 | / | constrained |
| 23 | CVPR | Heterogeneous Continual Learning | 0 | 0 | generators | discussed |
| 24 | CVPR | Decoupling Learning and Remembering: a Bilevel Memory Framework with Knowledge Projection for Task-Incremental Learning | 0 | 0 | model copies | compared |
| 25 | CVPR | Geometry and Uncertainty-Aware 3D Point Cloud Class-Incremental Semantic Segmentation | 0 | 0 | model copies | not discussed |
| 26 | CVPR | Continual Detection Transformer for Incremental Object Detection | 0.1 | 0.1 | model copies | not discussed |
| 27 | CVPR | Continual Semantic Segmentation with Automatic Memory Sample Selection | 0.001 | 0.01 | / | discussed |
| 28 | CVPR | Adaptive Plasticity Improvement for Continual Learning | 0 | 0 | gradient bases | compared |
| 29 | CVPR | VQACL: A Novel Visual Question Answering Continual Learning Setting | 0.001 | 0.09 | / | not discussed |
| 30 | CVPR | Task Difficulty Aware Parameter Allocation & Regularization for Lifelong Learning | 0 | 0 | model copies | discussed |
| 31 | CVPR | Computationally Budgeted Continual Learning: What Does Matter? | 1 | 1 | / | constrained |
| 32 | CVPR | CoMFormer: Continual Learning in Semantic and Panoptic Segmentation | 0 | 0 | model copies | not discussed |
| 33 | CVPR | PIVOT: Prompting for Video Continual Learning | 0.006 | 0.1 | / | not discussed |
| 34 | CVPR | Class-Incremental Exemplar Compression for Class-Incremental Learning | 0.003 | 0.02 | / | discussed |
| 35 | CVPR | PCR: Proxy-based Contrastive Replay for Online Class-Incremental Continual Learning | 0.002 | 0.1 | / | not discussed |
| 36 | CVPR | AttriCLIP: A Non-Incremental Learner for Incremental Knowledge Learning | 0 | 0 | / | not discussed |
| 37 | CVPR | Learning with Fantasy: Semantic-Aware Virtual Contrastive Constraint for Few-Shot Class-Incremental Learning | 0 | 0 | prototypes | not discussed |
| 38 | CVPR | On the Stability-Plasticity Dilemma of Class-Incremental Learning | 0.01 | 0.01 | / | discussed |
| 39 | CVPR | MetaMix: Towards Corruption-Robust Continual Learning with Temporally Self-Adaptive Data Transformation | 0.01 | 0.06 | / | compared |

**Table 1 continued from previous page**

| | | | | | | |
|---|---|---|---|---|---|---|
| 40 | CVPR | Exploring Data Geometry for Continual Learning | 0.004 | 0.04 | / | not discussed |
| 41 | NeurIPS | Uncertainty-Aware Hierarchical Refinement for Incremental Implicitly-Refined Classification | 0 | 0 | model copies | not discussed |
| 42 | NeurIPS | Learning a Condensed Frame for Memory-Efficient Video Class-Incremental Learning | 0.001 | 0.03 | / | not discussed |
| 43 | NeurIPS | S-Prompts Learning with Pre-trained Transformers: An Occam's Razor for Domain Incremental Learning | 0 | 0 | / | not discussed |
| 44 | NeurIPS | NOTE: Robust Continual Test-time Adaptation Against Temporal Correlation | 0.5 | 0.5 | / | discussed |
| 45 | NeurIPS | Decomposed Knowledge Distillation for Class-Incremental Semantic Segmentation | 0.008 | 0.1 | / | discussed |
| 46 | NeurIPS | Few-Shot Continual Active Learning by a Robot | 0 | 0 | prototypes | not discussed |
| 47 | NeurIPS | Navigating Memory Construction by Global Pseudo-Task Simulation for Continual Learning | 0.004 | 0.04 | / | compared |
| 48 | NeurIPS | SparCL: Sparse Continual Learning on the Edge | 0.004 | 0.01 | / | optimized |
| 49 | NeurIPS | A simple but strong baseline for online continual learning: Repeated Augmented Rehearsal | 0.01 | 0.1 | / | compared |
| 50 | NeurIPS | Lifelong Neural Predictive Coding: Learning Cumulatively Online without Forgetting | 0 | 0 | / | not discussed |
| 51 | NeurIPS | A Theoretical Study on Solving Continual Learning | 0.004 | 0.04 | / | discussed |
| 52 | NeurIPS | Beyond Not-Forgetting: Continual Learning with Backward Knowledge Transfer | 0 | 0 | gradient bases | compared |
| 53 | NeurIPS | Task-Free Continual Learning via Online Discrepancy Distance Learning | 0.04 | 0.2 | / | compared |
| 54 | NeurIPS | Disentangling Transfer in Continual Reinforcement Learning | 0.1 | 0.1 | / | not discussed |
| 55 | NeurIPS | Less-forgetting Multi-lingual Fine-tuning | 0.5 | 0.5 | / | not discussed |
| 56 | NeurIPS | Model-based Lifelong Reinforcement Learning with Bayesian Exploration | 1 | 1 | / | discussed |
| 57 | NeurIPS | ALIFE: Adaptive Logit Regularizer and Feature Replay for Incremental Semantic Segmentation | 0.01 | 0.04 | / | not discussed |
| 58 | NeurIPS | Retrospective Adversarial Replay for Continual Learning | 0.004 | 0.2 | / | constrained |
| 59 | NeurIPS | ACIL: Analytic Class-Incremental Learning with Absolute Memorization and Privacy Protection | 0 | 0 | / | not discussed |
| 60 | NeurIPS | Memory Efficient Continual Learning with Transformers | 0.1 | 0.16 | / | compared |
| 61 | ICML | Poisoning Generative Replay In Continual Learning To Promote Forgetting | 0 | 0 | generators | not discussed |
| 62 | ICML | Dualhsic: Hsic-Bottleneck And Alignment For Continual Learning | 0.01 | 0.1 | / | discussed |
| 63 | ICML | Adaptive Compositional Continual Meta-Learning | 0 | 0 | / | discussed |
| 64 | ICML | Ddgr: Continual Learning With Deep Diffusion-Based Generative Replay | 0 | 0 | generators | compared |
| 65 | ICML | Neuro-Symbolic Continual Learning: Knowledge, Reasoning Shortcuts And Concept Rehearsal | 0.02 | 0.045 | / | compared |
| 66 | ICML | Birt: Bio-Inspired Replay In Vision Transformers For Continual Learning | 0.003 | 0.04 | / | discussed |
| 67 | ICML | Optimizing Mode Connectivity For Class Incremental Learning | 0.01 | 0.04 | / | not discussed |
| 68 | ICML | Continual Learners Are Incremental Model Generalizers | 0 | 0 | extra modules | discussed |
| 69 | ICML | Learnability And Algorithm For Continual Learning | 0.02 | 0.04 | prototypes | not discussed |
| 70 | ICML | Do You Remember? Overcoming Catastrophic Forgetting For Fake Audio Detection | 0 | 0 | / | not discussed |
| 71 | ICML | Continual Vision-Language Representation Learning With Off-Diagonal Information | 0 | 0.16 | model copies | not discussed |
| 72 | ICML | Prototype-Sample Relation Distillation: Towards Replay-Free Continual Learning | 0 | 0.1 | model copies | not discussed |
| 73 | ICML | Parameter-Level Soft-Masking For Continual Learning | 0 | 0 | model copies | not discussed |
| 74 | ICML | Does Continual Learning Equally Forget All Parameters? | 0.002 | 0.05 | / | compared |
| 75 | ICML | Lifelong Language Pretraining With Distribution-Specialized Experts | 0 | 0 | extra modules | discussed |
| 76 | ICML | Continual Task Allocation In Meta-Policy Network Via Sparse Prompting | 0 | 0 | / | not discussed |
| 77 | ICML | Incdsi: Incrementally Updatable Document Retrieval | 1 | 1 | / | optimized |

