# OpenReview forum: "Continual Learning: Applications and the Road Forward"
_TMLR — Accepted by TMLR_

### Review · Reviewer_vbve · 2023-12-19

**Summary Of Contributions:**

This paper is a survey paper to discuss the current state and future direction of continual learning. This paper presents the necessity of the continual learning; the past discussion points, such as memory usage and computation costs; and the future open problems.

**Audience:**

Yes

**Broader Impact Concerns:**

Not much of an ethical concern paper.

**Claims And Evidence:**

No

**Requested Changes:**

see the weakness

**Strengths And Weaknesses:**

Strength:

This is a plainly written survey paper without too much detail, so this article is very accessible to any person with some knowledge of machine learning.

Weakness:

1. This paper shows some surveys in quantitative method, i.e. Figure 1. However, the background context of the past research is not well discussed. For instance, author wrote "strongly restrict memory and do not discuss computation cost". There are valid reasons of such setting. For example, continual learning is often deployment-purposed, so they restrain the training with future memory usage. Subsequently, such training will take less amount of data, so the computational cost could be less important because of the reduced data to process compared to the main training data. These pros and cons are not discussed, and only authors view is prevalent.

Unbiased survey is crucial in writing a survey paper.

2. Only very limited survey is done with three conference events: ECCV 22, NIPS 22, CVPR 23. Now, we are at the end of 2023, so there will be plenty more events to cover. Moreover, why not include ICCV, ICML, etc? I would say that the survey should be exhaustive to be a published paper in TMLR.

3. More indepth insight is needed. The key development of continual learning is not being covered in any technical depth. There is no trace of the loss function development, or problem setting evolution in the field of continual learning.

---

> ### Author Response · Authors · 2024-02-06
>
> We would like to thank the reviewer for their valuable feedback and time. We answer the raised questions below. We updated the text of the paper based on your feedback (orange and red text in the revised version) and fixed the spelling errors you found.
>
> \
> **Memory restricted settings are important too and limited memory as a proxy to limit the computational cost.** We do not intend to convey the message that memory restricted settings are not important, so we added clarifications in Section 2 to make this clearer early on in the paper. We write “most papers strongly restrict memory use and do not discuss computational cost”, as this is what our review of the selected papers revealed. Most papers are in the bottom left of Figure 1, which represents this setting. We agree that there are good use cases for restricted memory use, and mention so at the end of Section 2. Section 3.3 is about on-device learning and thus strongly memory constrained, and its importance is again acknowledged in Section 4.1. The point we want to make is that both are relevant settings, but that the compute constrained setting is currently less studied.
>
> We agree that there is a connection between the amount of data that is used and the computational cost for a model to converge. In some settings, this might be enough. However, it also has some difficulties. Limiting the amount of stored data does not *necessarily* limit the computational cost and algorithmic choices can have a large impact. For instance, when using replay, it is not uncommon to have as many old samples as new samples in a training mini batch. In such cases, the computational complexity is not determined by the number of stored samples, but by the amount of new data, which can still be large. When restricting memory to restrict the computational cost, it might confound the influence of both. One does not know whether a performance degradation in these settings is the result of e.g. overfitting on limited data, or because of not having converged as a result of limiting the computational cost. We included a discussion on this in Section 4.1.
>
> \
> **Not enough papers in the survey of Section 2.** The goal of Section 2 is to get a glimpse of the current state of field and the typical assumptions made by recent continual learning papers. We updated the initial results with extra papers from ICML ‘23 and added a color differentiation in Figure 1. The papers from ICML validate our findings with the earlier conferences, confirming that these ways of treating memory and computational cost are common in current continual learning literature and consistent over all four conferences. We do not try to list all existing papers that deviate from these common assumptions, but rather describe the general trends, which we believe are clear enough with these four conferences.
>
> The aim of our paper is not to survey the continual learning field, but to make a case for continual learning and why it is important to care about it from the perspective of solving real-world problems. We do not intend to offer a complete overview of the current state of the field. To avoid confusion, we no longer use the word survey in the updated version.
>
> \
> **No technical coverage.** Although interesting, this is beyond the scope of this paper. We wanted to make a case for why continual learning is a relevant research domain and why we need to keep working on things like the development of new and improved loss functions of continual learning. There are already comprehensive surveys out there that cover those aspects, e.g. [a-c]. We added this clarification to the end of the introduction.
>
> \
> [a] M. Masana, X. Liu, B. Twardowski, M. Menta, A. D. Bagdanov and J. van de Weijer, "Class-Incremental Learning: Survey and Performance Evaluation on Image Classification," in *IEEE Transactions on Pattern Analysis and Machine Intelligence*, vol. 45, no. 5, pp. 5513-5533, 1 May 2023, doi: 10.1109/TPAMI.2022.3213473
>
> [b] Zhou, D. W., Wang, Q. W., Qi, Z. H., Ye, H. J., Zhan, D. C., & Liu, Z. (2023). Deep class-incremental learning: A survey. *arXiv preprint arXiv:2302.03648.*
>
> [c] Wang, L., Zhang, X., Su, H., & Zhu, J. (2023). A comprehensive survey of continual learning: Theory, method and application. *arXiv preprint arXiv:2302.00487.*

---

### Review · Reviewer_A78Y · 2023-12-28

**Summary Of Contributions:**

This paper presents a brief survey of recent papers on continual learning (CL for short, a.k.a. lifelong learning), provides five motivating scenarios (error correction, personalization, on-device learning, model re-training and reinforcement learning) and finally identifies four possible future research directions (computational cost, theory, large-scale learning and open-world learning).

**Audience:**

Yes

**Claims And Evidence:**

No

**Requested Changes:**

(1) Include more papers from recent years and from more venues

(2) Provide a clear categorization

Minor comments

- 10.000 GPU -> 10,000 GPU
- Figure 1: why use dots when a number suffices. If visualization is preferred, one can use a heat map.
- make learning efficiently -> make learning efficient
- Citation: use \citet when the citation is part of the sentence and \citep otherwise. For example, in Sec.3.5, “in off policy RL settings Sutton & Barto (2018)” should be “in off policy RL settings (Sutton & Barto 2018)”

**Strengths And Weaknesses:**

Strengths

- Summarizing very recent papers and pointing out computational cost is much less explored in the literature.
- Writing is mostly clear and easy to read

Weaknesses

- The number of surveyed papers is limited. The surveyed papers come from only three conferences (ECCV 22, NeurIPS 22 and CVPR 23), which is far from enough for a survey paper. It would be important to cover other important venues such as ICML and ICLR.
- The categorization is unclear. As a survey paper, it is crucial to provide a clear categorization of existing methods, distinguishing the key differences and applicability of various algorithms, which is unfortunately missing for the current paper. For example, The title of Sec.3.1 is “Adapting machine learning models locally” but the content is more about correcting prior mistakes and there seems to be nothing specific about “locally”. Moreover, the abstract calls it “model-editing”, which overlaps with Sec.3.2 personalization as one may want to adjust the model for the purpose of personalization/adaptation.
- Some motivations are not well connected to continual learning. For example, Sec.3.2 is more related to transfer learning (one-time; adapting a model learned from vast data to a specific domain) than continual learning (lifelong). “Further, Internet scraped data often do not contain (enough) information to reach the best performance in specialized application domains like science and user sentiment analysis.” I beg to disagree. The problem here is more about identifying the relevant information than not having enough information. There are several similar claims that are not substantiated by papers or experiments.
- Only pointing out potential problems without providing enough solutions

---

> ### Author Response · Authors · 2024-02-06
>
> We thank the reviewer for their valuable feedback and time. Below we clarify the raised points. We updated the paper in several places in line with your comments (pink and red text in the updated version), as detailed below.
>
> \
> **Not enough papers in the survey of Section 2.** The goal of Section 2 is to get a glimpse of the current state of field and the typical assumptions made by recent continual learning papers. We updated the initial results with extra papers from ICML ‘23 and added a color differentiation in Figure 1. The papers from ICML validate our findings with the earlier conferences, confirming that these ways of treating memory and computational cost are common in current continual learning literature and consistent over all four conferences. We do not try to list all existing papers that deviate from these common assumptions, but rather describe the general trends, which we believe are clear enough with these four conferences.
>
> The aim of our paper is not to survey the continual learning field, but to make a case for continual learning and why it is important to care about it from the perspective of solving real-world problems. We do not intend to offer a complete overview of the current state of the field. To avoid confusion, we no longer use the word survey in the updated version.
>
> \
> **Clarity of the distinction between the problems in Section 3.** The words used in the paper to refer to the different subsections and their titles did not always align. We updated the titles and some of the text (e.g. the abstract) to let these match better and improve clarity. Although we no longer include it in the title of Section 3.1, we have added an explanation of what ‘local’ means in this context, i.e. changing the input output mapping of a model to fix a mistake should only alter the output to inputs in the direct neighborhood of this specific input. This setting is called model editing in previous literature, which is why we also called it as such (Mitchel et al. 2022). In Section 3.1, the number of required changes are rather small and originate from previously learned mistakes, or from outlier data that were not included in the original data. Different mistakes and outliers are often not strongly related (e.g. “khashoggi” and “Covid-19”), but are outliers of the original training data (e.g. recognizing a car with open doors). In contrast, for model specialization and personalization, the new training data are strongly related to each other (e.g. they all belong to the medical domain or are personal words belonging to a single user). We added a clarification in Section 3.2 to clarify this difference.
>
> Finally, we would like to say that we do not mean to introduce these problems as a strict categorization without any overlap, or to be exhaustive. These problems are intended as examples to illustrate the need for continual learning in real-world problems. There might be cases where some of the properties of model editing and personalization are both present. Yet sometimes there are clear differences, as in some of our examples, hence we chose to discuss them in separate subsections. We hope this answers your question, as we are not sure we fully understood the comment. If you have further comments, please let us know.
>
> \
> **On the connection to transfer learning.** There is a connection to transfer learning in Section 3.2, but an important difference is that, while transfer learning only cares about performance on the source domain, the problem setting in Section 3.2 additionally requires avoiding a performance decrease on the source domain. E.g. in Example 3.2, it is not enough to only learn the specialized medical words (target domain), a good model should still know how to transcribe common day-to-day words (source domain) to write down a medical conversation correctly. We discuss this in the second paragraph of Section 3.2: “However, these methods do not…” and added additional clarification to the last paragraph of Section 3.2.

---

> ### Author Response · Authors · 2024-02-06
> **Continuation of the previous**
>
> **Internet data could contain specialized data.** We updated our claim about internet scraped data to be more nuanced, as it might indeed contain the relevant information. We now only claim that internet scraped data do not always contain enough information. We believe this claim is backed up by  Beltagy et al. (2019). They constructed a vocabulary based on scientific papers rather than how WordPiece (used in e.g. Bert models) does this, which is based on the most frequently used words in common speech. They find an overlap of only 42%. We do agree that in some cases there might be enough data in the original training set already. If the performance on the specialized data (within the larger dataset) is not good enough, this requires more training on that specific bit of the data (given that we can extract it, which is not always straightforward). This is a shift in training data distribution, which when naively approached, can cause forgetting (this is also what Dingliwal et al. (2023) do in Example 3.2). We updated the text to reflect these ideas better.
>
> **About not providing potential solutions.** Although this certainly is interesting, we believe that this is beyond the scope of our paper. We hope that the insights we provide are helpful for developing new solutions in the future, as that is, indeed, the final goal.
>
> **Visualization of Figure 1.** Thank you for this suggestion. We added colors to distinguish the different conferences now, for which we think this way is the best way to visualize the data.

---

### Review · Reviewer_tRJw · 2024-01-24

**Summary Of Contributions:**

The paper argues that the field of continual learning is limiting itself to a few arbitrary use-cases, when in-fact continual learning has much wider applicability. The authors argue about multiple use-cases of CL that are ignored by most of the current research. These use-cases include (1) specializing the AI systems to their environment (as opposed to fixing them at the time of deployment), editing the knowledge of the systems when they make mistakes, (3) saving computation by not retraining models from scratch, (4) putting more emphasis on computational constraints, as opposed to memory constraints,  and tacking continual learning in a more principled way from the lens of theory.

**Audience:**

Yes

**Broader Impact Concerns:**

No comments

**Claims And Evidence:**

Yes

**Requested Changes:**

I would request the authors to read or skim the papers I listed, and support their arguments with the evidence in the papers (if possible). I'm not making any hard constraints, as the authors have already done an excellent job of supporting their arguments using existing literature, and I trust that they will do a good job including the papers I mentioned.

**Strengths And Weaknesses:**

## Strengths
This is a much needed paper for the field of continual learning. A large part of the CL community is plagued by a hand-crafted problem setting---memory constrained supervised learning---that only covers a small aspect of the need for continual learning, and this paper makes a clear and concise case for why CL has a much wider appeal.

I also wholly agree with the authors that the motivations used by the current supervised CL community---privacy and memory constrained---are not well thought out, and the meat of the need for CL lies elsewhere---adapting to distribution shifts, adding and removing knowledge, and specializing the models to their environment at the time of deployment, etc.

I would strongly recommend the paper to be accepted.

## Small gripes

1. I think the paper can emphasize some points better. More specifically, prior work by Silver, Koop, Müller, and Sutton has empirically shown that continually learning systems can out-perform fixed systems, laying a strong case for CL. A discussion on them would be useful. The two relevant papers are:

i). Sutton, Richard S., Anna Koop, and David Silver. "On the role of tracking in stationary environments." Proceedings of the 24th international conference on Machine learning. 2007.
ii) Silver, David, Richard S. Sutton, and Martin Müller. "Sample-based learning and search with permanent and transient memories." Proceedings of the 25th international conference on Machine learning. 2008.


2. There has been some work that empirically verifies the impact of computational constraints on online learning/lifelong learning systems. The following paper shows that under computational constraints of FLOPs-per-step, algorithms that are believed to be inferior can out-perform the popular algorithms (See Section 2 for problem setting and computational constraints):

Javed, K., Shah, H., Sutton, R. S., & White, M. (2023). Scalable Real-Time Recurrent Learning Using Columnar-Constructive Networks. Journal of Machine Learning Research, 24, 1-34.

3. "Freezing, or using non-uniform learning rates, might also be necessary when data is limited to prevent optimization and overfitting issues"

Using non-uniform learning rates can also help with better credit assignment for continual learning. Mainly, it can allow continual learning systems to learn to preserve some knowledge, and overwrite other. The following paper touches on these ideas:

Sutton, R. S. (1992, July). Adapting bias by gradient descent: An incremental version of delta-bar-delta. In AAAI (Vol. 92, pp. 171-176).

There are other works with similar ideas as well, such as Stochastic Meta-Descent (Schraudolph 1999).

---

> ### Author Response · Authors · 2024-02-06
>
> We thank the reviewer for their positive feedback, the useful suggestions and the interesting literature. These papers allowed us to look at some of the topics from a different angle and enrich the discussion. We added discussions and references at several places throughout the paper in response to the reviewer’s suggestions (blue text in the updated version).
>
> Summarized, in 4.1. we add the example that strictly limiting computational cost can alter the rank order of algorithms on a particular benchmark (Javed et al. 2023). In 4.2. we now discuss that by acknowledging the temporal coherence of the world, tracking solutions can outperform static ones (Sutton et al. 2007, Silver et al. 2008) and in 4.3 when discussing non-uniform learning rates we refer to Sutton (1992).

---

### Decision · Action_Editor_heM1 · 2024-03-20

**Recommendation:** Accept with minor revision

**Comment:**

The decision to accept the paper is based on its substantial contribution to the discourse on continual learning in machine learning. The authors effectively identify and challenge the current limitations within the field, advocating for a better perspective on the applications and implications of CL. Their engagement with recent literature and incorporation of feedback from reviewers have strengthened the paper’s arguments.

While the paper may not provide an exhaustive survey of the field it is well-positioned to stimulate further discussion and research in the field, making it a valuable addition to the journal.

I strongly recommend authors to further incorporate suggestions from reviews in their revision and extend the scope of the paper with the work published during the review process.

**Audience:**

TMLR's audience would be interested in the findings of this paper. The paper offers a sober and insightful perspective on continual learning. Its focus on understanding of CL and promoting a well designed setting is timely and has the potential to immensely influence future research directions.

**Claims And Evidence:**

The claims made in the submission are largely supported by accurate and convincing evidence. The authors have successfully engaged with the recent literature to support their arguments about the broader applicability and importance of continual learning (CL) in realistic settings. They effectively address the prevalent focus on memory-constraint within the CL community, presenting compelling arguments for why this focus is problematic.

However, reviewers point about the limited number of surveyed papers being insufficient for a comprehensive survey is valid. This limitation is mitigated by the authors' clarification that their goal is not to provide an exhaustive survey but to make a case for continual learning's relevance in real-world problems. Moreover, the authors responded to this concern by adding more papers from which strengthens their argument.

The lack of in-depth technical details, as pointed out by Reviewer vbve, is noted but it is outside the scope of this paper not impacting my decision.